# Inner Ear Pharmacotherapy for Residual Hearing Preservation in Cochlear Implant Surgery: A Systematic Review

**DOI:** 10.3390/biom12040529

**Published:** 2022-03-31

**Authors:** Quentin-Alexandre Parys, Pauline Van Bulck, Elke Loos, Nicolas Verhaert

**Affiliations:** 1Department of Otorhinolaryngology-Head and Neck Surgery, University Hospitals Leuven, 3000 Leuven, Belgium; quentin-alexandre.parys@uzleuven.be (Q.-A.P.); pauline.vanbulck@uzleuven.be (P.V.B.); elke.loos@kuleuven.be (E.L.); 2Department of Neurosciences, Research Group Experimental Oto-Rhino-Laryngology (ExpORL), KU Leuven, University of Leuven, 3000 Leuven, Belgium

**Keywords:** cochlear implant, drug delivery, inner ear, corticosteroids, dexamethasone, fibrosis, anti-inflammatory, residual hearing, hearing preservation

## Abstract

Cochlear implantation initiates an inflammatory cascade in which both acute insertion trauma and chronic foreign body reaction lead to intracochlear fibrosis and loss of residual hearing. Several strategies have been proposed to attenuate the local reactive process after implantation, including intracochlear drug delivery. The present study gives an overview of what is being investigated in the field of inner ear therapeutics and cochlear implant surgery. The aim is to evaluate its potential benefit in clinical practice. A systematic search was conducted in PubMed, Embase, and Cochrane Library databases identifying comparative prospective studies examining the effect of direct inner ear drug application on mechanical cochlear trauma. Both animal and human studies were considered and all studies were assessed for quality according to the validated risk of bias tools. Intracochlear administration of drugs is a feasible method to reduce the local inflammatory reaction following cochlear implantation. In animal studies, corticosteroid use had a significant effect on outcome measures including auditory brainstem response, impedance, and histological changes. This effect was, however, only durable with prolonged drug delivery. Significant differences in outcome were predominantly seen in studies where the cochlear damage was extensive. Six additional reports assessing non-steroidal agents were found. Overall, evidence of anti-inflammatory effects in humans is still scarce.

## 1. Introduction

Cochlear implant (CI) surgery has become indispensable for helping patients with severe to profound sensorineural hearing loss. Continuous technological evolution, increasing surgical experience, and the introduction of electric-acoustic stimulation have led to improved outcomes and thus a broadening of implantation criteria [1]. As a result, subjects with an increasing degree of residual hearing are being implanted worldwide. Inner ear surgery and electrode array insertion initiate an inflammatory cascade in which both acute insertion trauma and chronic foreign body reaction lead to intracochlear fibrosis and progressive loss of residual hearing [2,3]. Since residual hearing preservation is known to positively affect speech perception in CI patients [4], attempts are made to attenuate this reactive process after implantation. Potential strategies would certainly benefit those requiring multiple implantations throughout their lives. Current research focuses on advances in electrode design and surgical technique, perioperative monitoring of the cochlear function, and the use of anti-inflammatory drugs. Lately, the latter has become a major point of interest in the literature.

So far, three modes of drug delivery have been investigated: systemic, intratympanic, and intracochlear. Systemic delivery, although straightforward, suffers greatly from the ‘first pass’ effect and blood-labyrinth barrier necessitating elevated drug concentrations, which, together with the need for prolonged administration, pose a risk for adverse effects [5]. This hurdle is surpassed with intratympanic infiltration [6]. Nevertheless, slow and incomplete diffusion through the round window membrane results in inconsistent perilymph concentrations and prevents sufficient accumulation of drugs for most therapeutical purposes [6,7,8]. This is especially the case for middle and apical regions of the cochlea representing lower frequencies involved in residual hearing. To overcome this problem, efforts have been made to ensure sustained and more predictable release into the scala tympani after intratympanic delivery [9]. Aiming for greater and more consistent drug levels, intracochlear delivery systems have emerged. Being unconstrained by the round window, direct administration of drugs into the scala tympani extends the therapeutic range. This is achieved by reaching higher peak concentrations faster and more steadily and by reducing the basal to apical drug gradient [10]. Besides direct injections, osmotic pumps and drug-eluting electrodes explore the feasibility of prolonged intrascalar delivery [9]. Most studies assess perioperative topical corticosteroid (CS) use because of its well-documented modulating effect on cochlear inflammation [11]. In addition to CS, non-steroidal drugs with anti-inflammatory properties, substances that promote neural health and/or growth, and therapies that focus on minimizing local tissue remodeling are being investigated in conjunction with CI surgery.

The heterogeneity between studies regarding trauma and implant model, choice and dosage of the drug, period of follow up, outcome, etc., prevent pertinent conclusions from being drawn. The purpose of the present study is to give an overview of what is being investigated nowadays in the field of inner ear therapeutics and CI surgery and to evaluate its potential benefit when it comes to prevention of tissue response, protection of neural health, and the preservation of residual hearing. By performing a systematic review of the literature, we aim to set the path for future clinical guidelines in cochlear implantation surgery.

## 2. Materials and Methods

### 2.1. Search Strategy & Study Selection

We conducted a systematic search of all the English literature published before 1 January 2021 following the PRISMA guidelines. The identification stage entailed searching the online databases PubMed, Embase, and Cochrane library using the following keywords and related keywords: “Cochlear implantation”, “Fibrosis”, “Inflammation” and “Hearing preservation” (Appendix A). Pharmaceutical interventions were purposely excluded from the search query to obtain an exhaustive search. Additionally, reference lists of all included studies were assessed for other relevant articles. 

The entire inclusion process was carried out by two reviewers independently (QAP & PVB) with the use of Rayyan software for systematic reviews [12]. The following inclusion criteria were applied: (1) prospective study design with a matching control group (*n* ≥ 5 per group); (2) involving a mechanical insult to the cochlea; (3) where therapy is directly delivered into the inner ear; (4) with a clear description of outcome measures. Both animal and human studies were considered. Non-English articles, in vitro studies, drug dilution studies, review articles, and articles published before 2000 were excluded. Subgroups within included studies not fitting the inclusion criteria were not covered in this review.

### 2.2. Outcome Measures

Following outcome measures were taken into account: impedance, auditory brainstem response (ABR), compound action potential (CAP), otoacoustic emissions (OAE), hair cell (HC) counting, spiral ganglion neuron (SGN) density, and signs of tissue response (fibrosis, osteogenesis, inflammatory cell count, cytokine detection,…). Additional information regarding the technical outcome measures is provided in Appendix B.

### 2.3. Quality Assessment

Study quality and the risk of bias were assessed by two researchers independently (QAP & PVB) using Risk of Bias tools. Animal studies were evaluated according to the “SYRCLE’s risk of bias tool for animal intervention studies” (Appendix A) [13] and human studies according to the “Checklist for the assessment of the methodological quality” (Appendix A) [14]. Disagreements between the reviewers were resolved through discussion.

## 3. Results

### 3.1. Systematic Search

A total number of 3911 articles were retrieved using the systematic search described above. Titles and abstracts of 2282 articles were screened after duplicate records were deleted. Upon the inclusion and exclusion criteria, 45 articles remained for eligibility evaluation and thorough full-text review. In the absence of a clear method description or the case of insufficient data, the author of that manuscript was contacted. Reference searching did not result in the inclusion of additional records, indicating the exhaustiveness of the search strategy. Finally, 26 relevant papers were included in this review (Figure 1). Study characteristics of the included reports are depicted in Table 1. Some studies were conducted by the same research group or performed in the same laboratory. In total, 11 different institutions are represented in this review. The entire inclusion process was done by two researchers independently (QAP and PVB).

### 3.2. Quality Assessment

Quality assessment was performed separately for animal and human studies as shown in Table 2 and Table 3. Manuscripts of animal studies did not provide information on how randomization occurred and only a handful disclosed whether outcome assessors were blinded. Moreover, in two studies, all left ears were implanted while right ears served as unoperated controls. Thus, the results of these studies are subjected to a moderate to high risk of selection and detection bias [13]. Performance bias was not assessed because no study provided information in this regard and it was not deemed relevant for this review. The four human studies were graded as fair quality according to the risk of bias tool of Downs and Black [14]. Here, similarly, a lack of reporting on randomization and blinding of investigators leads to an increased risk of bias. Since only studies with a prospective study design were included, the risk of publication bias is limited.

### 3.3. Pharmaceutical Interventions

#### 3.3.1. Glucocorticoids

Of the 26 included studies, 20 assessed the effectiveness of locally administered glucocorticoids on hearing preservation following cochlear implantation. Three methods were used for drug delivery: intracochlear injection, osmotic pump via intracochlear microcatheter, and drug-eluting electrodes (DEE). A summary of the main findings per study is added in Appendix A.

##### Intracochlear Injection

The effect of a one-shot intracochlear injection of triamcinolone and dexamethasone (DEX) was studied by Huang et al. in an animal model [15]. They found no significant differences regarding impedance changes, the extent of fibrous tissue growth, and reduction of SGN densities between experimental and control groups at any point during the five-month follow-up period. Another study compared the same corticosteroids with the injection of artificial perilymph [16]. Significantly lower CAP threshold shifts after implantation were observed using dexamethasone (*p* < 0.05). This effect was significant at the mid-frequencies during the first week and was lost thereafter. Triamcinolone-treated ears, on the other hand, showed higher initial shifts post-implantation, which decreased after the first week and ended up being significantly lower by day 90 (*p* < 0.05). The differences were marked at the low- and mid-frequencies. No reduction of the tissue growth at the scala tympani was seen in the treatment groups, nor was there a correlation between the amount of tissue growth within the cochlea and hearing loss in this study. Similarly, Lyu et al. found that slow intracochlear injection resulted in lower threshold shifts at all measured frequencies 2 months after surgery (*p* < 0.05) [17]. Additionally, lower concentrations of pro-inflammatory cytokines (IL-1β, IL-6, TNF- α, and NOS2), better hair cell survival, and less fibrosis were observed in the intracochlear group compared to untreated controls. Paasche et al. performed a study on human subjects evaluating the benefit of intracochlear injection of triamcinolone prior to cochlear implant insertion. Measured impedances remained significantly lower in the steroid group compared to the control group on days 3 to 10 (*p* < 0.01) and on weeks 2 and 3 (*p* < 0.05) but increased thereafter [18]. Mean impedances in the control group were higher at the basal region (6.84 kΩ) compared to the apical region (5.94 kΩ), whereas the opposite was true for the treatment group (2.21 kΩ and 2.91 kΩ, respectively). In the same human study population Paasche et al. subsequently investigated the long-term effects of one-off steroid deposition and demonstrated that, although the effect attenuated over time, impedances remained lower for an extended time of up to three to four years after implantation [19]. These findings did not reach significance. Additionally, in human subjects, Prenzler et al. used a 20 mm intracochlear microcatheter to administer triamcinolone deeper into the cochlea before surgery aiming to extend the potential effect of steroids towards the apex [20]. After implantation, mean impedance started to rise in the control group but not in the catheter group, reaching a significant effect on day 10 (*p* < 0.05). However, when subdividing into basal, middle, and apical contacts, only significant differences were observed in the middle electrodes (*p* < 0.0001 at day 10 and *p* < 0.001 at day 17). Over time, impedance increased in the steroid group while staying relatively stable in the control group, diminishing the effect of steroids after three weeks. Electrical stimulation reduced impedance in both groups at first fitting and at three months.

##### Intracochlear Microcatheter

Three studies inserted an intracochlear microcatheter peri-operatively. To achieve prolonged delivery of steroids, however, the catheter was kept in place and connected to an osmotic minipump. In doing so, Eshraghi et al. demonstrated no significant differences in median ABR thresholds at any frequency between the experimental cochleae receiving dexamethasone and the unoperated ears at day 30 post-electrode insertion trauma (EIT) (*p* > 0.05), whereas early recordings after EIT were significantly worse in the experimental ears at all frequencies but 0.5 kHz [21]. Moreover, in the EIT-only cochleae, significant differences remained at 4 kHz and 16 kHz throughout the entire follow-up period of 30 days. The same research group obtained similar results using an organic soluble form of dexamethasone, making it suitable for use in polymer-coated drug-eluting electrode arrays [22]. In contrast, Scheper et al. noticed higher initial ABR thresholds in the DEX group compared to controls (*p* < 0.05) immediately after implantation in deafened guinea pigs [23]. The mean threshold decreased thereafter in the treatment group and increased in the control group to reach a similar level at 4 weeks. In this same study, the median SGN survival at the basal region was significantly greater in the DEX group receiving electrical stimulation compared to the other groups not receiving electrical stimulation, not receiving DEX, and cochleae only receiving artificial perilymph (*p* < 0.001). When taking the entire cochlear length into account only the latter two groups had significantly reduced SGN numbers (*p* < 0.01). No correlation between the ABR threshold and SGN number was seen. 

##### Drug-Eluting Electrode

Twelve studies investigated drug-eluting electrodes containing corticosteroids. Only one was performed on human subjects. Farhadi et al. examined inflammatory responses following cochlear implantation with DEX-loaded electrodes and reported a significant reduction in fibrocyte, macrophage, and giant cell infiltration at day 3 as well as lymphocyte, macrophage infiltration, and capillary formation at day 13 [24]. Fibrotic tissue formation was not significantly different between the subgroups in a study by Liu et al., yet statistically significant hearing recovery was seen on ABR from week 1 to 12 postoperatively at frequencies 8–24 kHz (*p* < 0.05) and on DPOAE from 8–16 Hz (*p* < 0.05) [25]. Furthermore, intergroup differences in ABR recordings became more apparent after 3 weeks. In line with these findings, Douchement et al. confirmed preservation of residual hearing on ABR at 500 Hz, 1 kHz, 4 kHz, and 16 kHz (*p* < 0.05) 4–6 weeks after implantation with effects persisting at 16 kHz after one year (*p* = 0.0103) [26]. At 500 Hz and 1 kHz, on the other hand, significantly worse ABR thresholds were observed after one year (*p* = 0.0010 and *p* = 0.0368, respectively). A study by Bas et al. demonstrated preservation of hearing with both 1% and 10% DEX-eluting arrays. By postoperative day 90, threshold shifts reached pre-EIT levels at all frequencies tested [27]. Although no significant differences were observed between these two concentrations, their otoprotective effect was greater than DEX 0% (*p* < 0.001 for all frequencies) and DEX 0.1% indicating a potential dose-dependent relationship. The same results were obtained by Astolfi et al. [28]. These researchers noted a progressive recovery of the initial threshold shift at all frequencies in the drug-eluting group. In non-eluting controls, similar postoperative threshold shifts were measured but hearing worsened over time. 

DEX treatment did not significantly affect hearing preservation or tissue formation in other reports [29,30,31]. Scheper et al. looked at SGN survival and concluded that dexamethasone alone did not affect SGN density [23]. Threshold shifts in the DEX treatment group were greater at 16 kHz before and at both 1 and 16 kHz after electrical stimulation compared to non-eluting controls (*p* < 0.05) [31]. On the other hand, impedance measures were significantly lower in dexamethasone-treated groups. Weekly electrical stimulation mitigated impedance changes in all groups. Fibrous tissue growth followed the same pattern, having a positive correlation with impedance but not with hearing loss. 

Work by Chambers et al. demonstrated greater hearing threshold shifts in implantation models eliciting more mechanical trauma regardless of DEX [32]. This study examined the benefit of DEX in two implant models: a low trauma model where an electrode array was carefully introduced into the cochlea, and a high trauma model where a steel wire was inserted and withdrawn twice prior to insertion of the electrode array. At 4 weeks post-implantation, significant differences in thresholds at 16, 24, and 32 kHz were observed between the low and high trauma models (*p* < 0.05). The DEE in the high trauma model did not yield better results than the control electrodes of the same trauma model. Histologically, however, significantly more new bone formation was present in the latter compared to the other two groups (low trauma group, *p* < 0.001; high trauma DEX-group, *p* = 0.025). Furthermore, DEX had a positive effect on SGN survival in the lower basal turn in the high trauma groups (*p* = 0.041). 

Two types of impedance measures were recorded following cochlear implantation by Needham et al. [33]. Mean monopolar impedance measures, representing the device power requirements, were higher in the DEX-eluting group immediately at implantation (*p* = 0.002) and stayed greater in comparison with controls during the 4 week observation period (*p* = 0.001). Four-point impedance measures on the other hand provide information about the local environment and decreased in the DEX-eluting array group relative to the standard array attaining significant differences over time (*p* < 0.001). 

One study examined DEX-eluting electrode arrays in human subjects [34]. For monopolar impedance measures, the group mean averaged across all electrodes and time points was significantly lower in the DEE group than in controls (mean values of 5.6 and 8.9 kΩ, respectively, *p* < 0.0001). With four-point impedance, a significant three-way interaction between the effect of device, time, and region was observed (*p* = 0.008). Impedance was significantly greater at 6, 12, and 24 months at the basal region (*p* < 0.001) and at 12 and 24 months at electrodes in the middle third of the array (*p* < 0.01). No significant differences were noted at the apical region (*p* > 0.05). For the control group, impedance increased significantly in the basal region between three and six months post-activation (*p* = 0.002). Hearing preservation was not assessed because of the insufficient residual hearing present. 

#### 3.3.2. Non-Glucocorticoids

The systematic search yielded six additional studies investigating six different non-steroidal inner ear therapeutics in animal cochleae. D-JNKI-1, a kinase inhibitor with a proven otoprotective effect, was administered into the cochlea using an osmotic pump for 7 days post-electrode insertion trauma [35]. This resulted in a decrease in ABR threshold changes (7.1 ± 7.1 dB vs. 29.2 ± 13.0 dB, *p* < 0.0001) and DPOAE amplitudes (−4.3 ± 3.6 vs. −18.7 ± 7.5, *p* < 0.0001) compared to EIT-only controls, predominantly due to less gradual loss of auditory function.

Another possible strategy to prevent fibrous tissue formation is the inhibition of proinflammatory cytokines and immune response mediators present in the inner ear. Ihler and colleagues used Etanercept, an inhibitor of tumor necrosis factor-alpha (TNF-α), in their model of cochlear insertion trauma [36]. Hearing thresholds were consistently lower in the treatment group on day 28 compared to controls reaching significance at 2 kHz, 4 kHz, and 8 kHz (*p* < 0.01). An increasing effect of drug delivery on hearing preservation was noted in the lower frequencies with a significant recovery only at 2 kHz at day 28 (*p* = 0.003). A third study group received artificial perilymph via the same delivery technique. Here, thresholds were generally lower than in controls and greater than those in the treatment group. None of these differences were significant.

We found two studies examining the potential of growth factors (GF) in hearing preservation. The first involved a DEE coated with hydrogels containing insulin-like growth factor 1 and/or hepatocyte growth factor [37]. The hearing loss inflicted by electrode insertion recovered more quickly in the subgroups receiving GF leading to a significantly lower threshold at all frequencies measured at the 4-week completion date (*p* < 0.05). Hydrogel-coated electrodes not containing GF showed a significant effect as well but only at average ABR (*p* < 0.05). No significant differences in HC survival or SGC density were observed in this study. The second study entailed brain-derived neurotrophic factor (BDNF)-producing mesenchymal stem cells (MSCs) administered through coated electrode arrays and/or intracochlear injection [38]. No benefit of MSC injection was found at the higher frequencies when inserting an electrode array in normal hearing cochleae. MSCs delivery did not lead to significant differences in impedance nor in the extent of fibrosis, yet deafened cochleae inserted with MSC-coated electrodes had significantly increased SGN densities across the cochlea compared to untreated deafened ears (*p* < 0.05). This effect was more prominent when focusing solely on the basal region (*p* < 0.001). Here, it appeared that implantation alone already resulted in better SGN preservation compared to no intervention in deafened ears (*p* < 0.01). No additional benefit was observed with infiltration of MSCs before implantation, even when assessing subregions of the cochlea separately.

As laminin, a component of the extracellular matrix, is known to regulate pro-inflammatory Schwann cells, Bas et al. theorized that laminin-coated electrodes may contribute to a more SGN-friendly environment [39]. Histological studies of cochleae at 10 days or 4 weeks post-implantation showed a significantly larger number of SGNs in the experimental group compared to uncoated controls (*p* < 0.05), with numbers comparable to unoperated contralateral cochleae (*p* > 0.05). Additionally, cochleae exposed to laminin showed SGN neurite processes projecting into the scala tympani at 4 weeks post-implantation. While acoustic ABR thresholds shifts decreased significantly at 8, 16 and 32 kHz (*p* = 0.0076, *p* = 0.0104 and *p* < 0.0035, respectively), and electric ABR recordings remained lower in the treatment group (*p* = 0.0006), impedances increased significantly in laminin-coated electrodes compared to controls (*p* < 0.0001). Lastly, a research group investigated whether a fibrinolytic agent, tissue-type plasminogen activator (tPA), would impact cochlear inflammation [40]. Animals were divided into low or high trauma groups, receiving inner ear irrigations of either tPA or saline immediately after implantation. At two weeks, tPA did not affect hearing significantly in any trauma model. A 4.2% reduction in tissue response was detected in the low trauma group with tPA infusion (*p* = 0.039). This effect was not present in the high trauma model (*p* = 0.950). 

## 4. Discussion

Differing strategies have emerged over the last decades in efforts to attenuate cochlear inflammation post-implantation. Here we looked only at inner ear therapeutics with anti-inflammatory and anti-apoptotic properties. Since seemingly countless pro-inflammatory factors have been identified, potential strategies are abundant. Corticosteroids were used most often. Dexamethasone and triamcinolone were the CS of choice due to their superior potency compared to other steroids. Dexamethasone is more potent than triamcinolone but is eliminated from the perilymph more rapidly [41]. Six protocols comprised intracochlear injections of CS, three in animal models and three in humans. A significant effect of treatment was reported in four of these studies on various outcomes including ABR, impedance, and histological changes, however, in all but one study this effect was temporary, not extending beyond several weeks. The follow-up period for that particular study was 2 months [17]. Prolonged delivery was achieved by an osmotic pump system with beneficial effects on EIT-induced hearing loss in normal-hearing guinea pigs [21,22]. In deafened guinea pigs, DEX and electrical stimulation had a synergetic effect on ABR threshold changes and SGN survival [23]. This outcome was limited to the basal region where DEX was administered. The neuroprotective effect of chronic stimulation has been described in the literature [42] and was also reported in other studies discussed in this review [19,20,31,33], but by mimicking the clinical situation in CI patients, the added value of DEX was demonstrated in a preclinical model. To extend the therapeutic effect over the entire cochlea, drug-eluting electrodes seem to be a more complete and feasible delivery method. In general, in this setting, CS appeared to reduce the inflammatory reaction after implantation, lower impedance, and help recover initial thresholds induced by EIT. Moreover, some studies indicate that the progressive increase in thresholds observed in the weeks following implantation can be prevented with CS elution. No difference were observed when comparing DEX 1% and DEX 10% [26,27,31]. In contrast, other studies reported a stronger increase in thresholds immediately after DEE-insertion [17,21,31]. A possible explanation for this observation is that the physical loading of electrode arrays results in more rigid, heavier, and more bulky devices causing more alterations to intracochlear structures during insertion [31,43]. In several reports, DEE did not affect hearing preservation and fibrosis. This was the case, for example, in an atraumatic animal model where untreated controls experienced little hearing loss, hence an additional effect of DEX could not be pointed out [29]. The great interstudy variation in outcome seen in this review could be explained, at least in part, by the great disparity in trauma inflected upon the cochlea during insertion. It is believed that CS use is effective provided that a certain level of trauma has occurred [27,29,30,31,32]. Since insertion trauma predominantly affects the basal region of the cochlea, high-frequency regions will benefit most from therapy. Differences in outcomes within one study could be interpreted in the same way, as various variables can influence the extent of cochlear damage. 

In addition to steroids, our search yielded six alternative therapeutical approaches with otoprotective properties. The mechanism of action of these therapeutics included prevention of apoptosis [35,37,38,39], inhibition of pro-inflammatory cytokines [36], stimulation of neurite growth [37,38,39], and breakdown of intracochlear thrombi [40]. In one of the studies, living stem cells producing and releasing an SGN protective factor, BDNF, was applied in order to potentially enable lifelong therapy. Although some studies demonstrate riveting results, the lack of replication makes it difficult to draw conclusions at this point. Nevertheless, alternative approaches to corticosteroids will most likely have a place in CI surgery since targeted interactions with inflammatory mediators can be elicited. Studies in which steroidal and non-steroidal drugs are administered together are needed to investigate their clinical complementarity.

Intracochlear delivery of drugs is believed to be safe. No major complications or adverse effects were reported in the included studies. Prolonged administration of CS leads to an increased risk of bacterial infection. Wei et al. demonstrated that cochlear damage caused by EIT increases the possibility of bacterial migration towards the meninges [44]. Microbiology analysis, however, could not find signs of bacterial presence on implanted DEX-eluting electrode arrays [28]. Thus, it remains uncertain whether local CS administration influences the risk of bacterial meningitis, but from the evidence presented no clear findings point in that direction. Multiple studies concluded that DEX was not toxic to SGN [23,29,33].

Wilk and colleagues reported a correlation between histological changes and impedance measures post-implantation [31]. These findings were also noted by Bas et al. [27]. Considering electrode impedance measurements are part of the clinical practice during the fitting of CI patients, information about the intracochlear environment of the recipient is readily available. No apparent correlation was observed between ABR thresholds and histological findings [23,31,32]. 

The following limitations should be taken into account when reading this review. A summary of the main limitations per study is added in Appendix A. First, human studies assessing inner ear therapeutics are scarce. All but four included studies used animal models, of which two were conducted on the same human subjects. Guinea pigs are the favored animals because their cochlear anatomy resembles those of humans. Second, while all studies shared similar study designs, heterogeneity was present in various aspects. Hearing at baseline ranged from normal to purposefully deafened ears in animal models and from human individuals with residual hearing to completely deaf. Additionally, the implant model was inconsistent. In multiple studies, EIT was mimicked by performing a cochleostomy followed by insertion and withdrawal of a steel electrode. Here, cochlear trauma is reduced to only the acute mechanical event which is only half the story when it comes to tissue response. Actual long-term implantation has to take place to investigate the influence of local drugs on the chronic foreign body reaction. Furthermore, the duration of follow-up was equally discrepant ranging from 7 days to 4 years with an average of 144.2 days. The lack of adequate follow-up was even more apparent when looking at studies that included histological analysis. Here, the average follow-up period was just 53.5 days. By that time, the process of inflammatory cell migration, adhesion, and proliferation may not be fully developed. Together, the lack of homogeneity made it impracticable to perform quantitative analysis of included studies. Third, drug release from eluting electrodes is finite. The follow-up time of included studies was not sufficient to assess this potential issue. Additionally, in experiments that examined tissue response, only several animals from each study group were sacrificed. This may result in low statistical power and overestimation of effect size. Finally, the quality assessment revealed a great risk of bias. Particularly selection and detection bias should be taken into account when reading this review.

## 5. Conclusions

This up-to-date systematic review of animal and human studies demonstrates that intracochlear administration of corticosteroid and non-corticosteroid agents is a safe and effective method to attenuate local tissue response following cochlear implant surgery. To achieve durable results, prolonged delivery is necessary. This can be realized appropriately using drug-eluting electrodes. More benefit of therapy is to be expected with greater electrode insertion trauma and at higher frequencies. However, data on the long-term effect of foreign body introduction in the cochlea is lacking and only a few studies have been conducted in humans concerning this matter. Cochlear damage and the loss of residual hearing after cochlear implantation are multifactorial and therefore complicated to foresee and prevent, a finding which was reflected in the high degree of variability within study groups. In addition, histological analysis suggests that direct inner ear therapy is not harmful to local structures and does not increase the risk of infection. Thus, we can conclude that intracochlear therapy will likely play an essential role in an integrated approach to minimize local tissue reaction after cochlear implantation. Besides preservation of residual acoustic hearing, prevention of local scarring will improve frequency resolution and therefore result in enhanced sound perception. Further research should be directed towards prospective human studies with adequate follow-up periods to correctly assess the long-term benefits of intracochlear therapeutics.

## Figures and Tables

**Figure 1 biomolecules-12-00529-f001:**
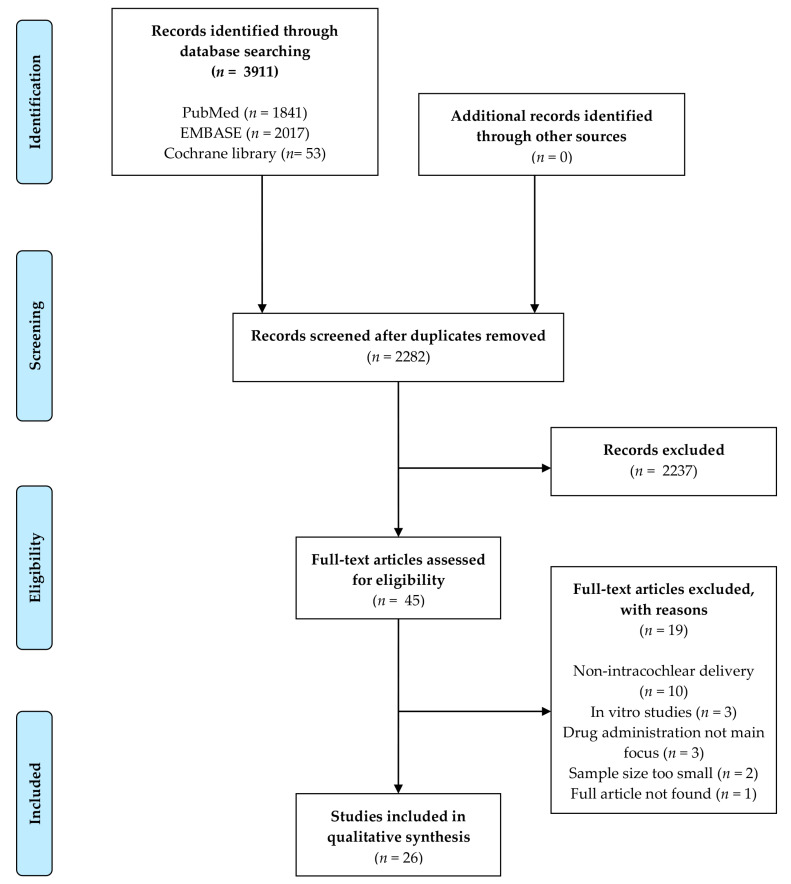
PRISMA Flow Diagram for Study Selection.

**Table 1 biomolecules-12-00529-t001:** Study characteristics.

Author, Year	Subjects	Intracochlear Delivery	Drug	Control	Total Ears	Follow-Up Duration	Primary Endpoint (s)
Huang, 2007	Guinea pigs, Cats	Injection	DEX, TRIAM	Multiple groups	*n* = 47	5 m	Impedance, histology
Braun, 2011	Guinea pigs	Injection	DEX, TRIAM	No injection; AP injection	*n* = 30	90 d	CAP, histology
Lyu, 2018	Guinea pigs	Injection	DEX	No injection	*n* = 180	60 d	ABR, histology
Paasche, 2006	Humans	Injection	TRIAM	No injection	*n* = 13	35 d	Impedance
Paasche, 2009	Humans	Injection	TRIAM	No injection	*n* = 15	≥3 y	Impedance
Prenzler, 2018	Humans	Injection	TRIAM	No injection	*n* = 10	90 d	Impedance
Eshraghi, 2007	Guinea pigs	Osmotic pump	DEX	AP-pump	*n* = 28	30 d	ABR
Vivero, 2008	Guinea pigs	Osmotic pump	DEX	No pump; AP pump	*n* = 88	30 d	ABR, HC counting
Scheper, 2017	Guinea pigs	Multiple methods	DEX	Multiple groups	*n* = 24	27 d	ABR, SGN survival
Farhadi, 2013	Guinea pigs	Drug-eluting electrode	DEX	Non-eluting electrode	*n* = 30	13 d	Histology
Liu, 2015	Guinea pigs	Drug-eluting electrode	DEX	Non-eluting electrode	*n* = 35	6 m	ABR, OAE, histology
Douchement, 2015	Gerbils	Drug-eluting electrode	DEX	Non-eluting electrode	*n* = 48	1 y	ABR
Bas, 2016	Guinea pigs	Drug-eluting electrode	DEX	Non-eluting electrode	*n* = ?	90 d	ABR, impedance, histology
Astolfi, 2016	Guinea pigs	Drug-eluting electrode	DEX	Non-eluting electrode	*n* = 32	14 d	CAP, histology
Ahmadi, 2019	Guinea pigs	Drug-eluting electrode	DEX	Non-eluting electrode	*n* = 20	4 m	ABR, impedance, histology
Stathopoulos, 2014	Guinea pigs	Drug-eluting electrode	DEX	Non-eluting electrode	*n* = 66	90 d	ABR, histology
Wilk, 2016	Guinea pigs	Drug-eluting electrode	DEX	Non-eluting electrode	*n* = 33	91 d	ABR, impedance, histology
Chambers, 2019	Guinea pigs	Drug-eluting electrode	DEX	Non-eluting electrode	*n* = 24	28 d	ABR, histology
Needham, 2020	Guinea pigs	Drug-eluting electrode	DEX	Non-eluting electrode	*n* = 50	40 d	Impedance, histology
Briggs, 2020	Humans	Drug-eluting electrode	DEX	Non-eluting electrode	*n* = 24	2 y	Impedance
Eshraghi, 2006	Guinea pigs	Osmotic pump	D-JNKI-1	Multiple groups	*n* = 37	7 d	ABR, OAE, HC counting
Ihler, 2014	Guinea pigs	Osmotic pump	Etanercept	No pump; AP pump	*n* = 15	28 d	ABR
Kikkawa, 2014	Guinea pigs	Drug-eluting electrode	IGF1, HGF or IGF1 + HGF	Multiple groups	*n* = 25	28 d	ABR, histology
Scheper, 2019	Guinea pigs	Multiple methods	MSCs	Multiple groups	*n* = 30	28 d	ABR, impedance, histology
Bas, 2019	Rats	Drug-eluting electrode	Laminin	Non-eluting electrode	*n* = 20	28 d	ABR, impedance, histology
Choong, 2019	Guinea pigs	Injection	tPA	Saline	*n* = 21	14 d	ABR, histology

DEX: dexamethasone; TRIAM: triamcinolone; IGF1: insulin-like growth factor 1; HGF: hepatocyte growth factor; MSCs: mesenchymal stem cells; tPA: tissue plasminogen activator; AP: artificial perilymph; ABR: auditory brainstem response; CAP: compound action potential; HC: hair cell; SGN: spiral ganglion neuron; OAE: otoacoustic emissions; d: days; m: months; y: years. Note that study groups not meeting the inclusion criteria were not included in this table.

**Table 2 biomolecules-12-00529-t002:** SYRCLE’s risk of bias tool for animal studies.

Study	Sequence Generation (Selection Bias)	Baseline Characterisitcs (Selection Bias)	Allocation Concealment (Selection Bias)	Random Housing (Performance Bias)	Blinding (Performance Bias)	Random Outcome Assessment (Detection Bias)	Blinding of Outcome Assessment (Detection Bias)	Incomplete Outcome Data (Attition Bias)	Selective Outcome Reporting (Reporting Bias)	Other Sources of Bias (Other)
Eshraghi, 2006	+	+	?	NR	NR	?	?	+	+	?
Huang, 2007	+	?	-	NR	NR	?	-	+	+	?
Eshragi, 2007	+	?	?	NR	NR	+	?	+	+	?
Vivero, 2008	+	?	?	NR	NR	?	?	+	+	-
Braun, 2011	?	?	?	NR	NR	?	?	+	+	?
Farhadi, 2013	+	+	?	NR	NR	+	?	+	+	?
Stathopoulos, 2014	?	?	?	NR	NR	?	?	+	+	?
Kikkawa, 2014	?	+	?	NR	NR	?	+	+	+	?
Ihler, 2014	+	+	?	NR	NR	?	?	+	+	?
Douchement, 2015	+	+	+	NR	NR	?	?	?	+	?
Liu, 2015	?	?	?	NR	NR	?	?	+	+	?
Wilk, 2016	+	+	?	NR	NR	?	?	+	+	?
Bas, 2016	+	+	?	NR	NR	?	?	+	+	-
Astolfi, 2016	+	+	?	NR	NR	?	?	+	+	?
Scheper, 2017	+	+	+	NR	NR	?	?	+	+	?
Lyu, 2018	?	?	?	NR	NR	+	+	+	+	-
Chambers, 2019	+	+	?	NR	NR	?	?	+	+	?
Scheper, 2019	+	+	+	NR	NR	+	+	+	+	?
Choong, 2019	+	+	+	NR	NR	+	+	+	+	?
Bas, 2019	?	?	?	NR	NR	?	?	+	+	?
Ahmadi, 2019	+	+	?	NR	NR	+	+	+	+	-
Needham, 2020	+	+	-	NR	NR	?	-	+	+	?

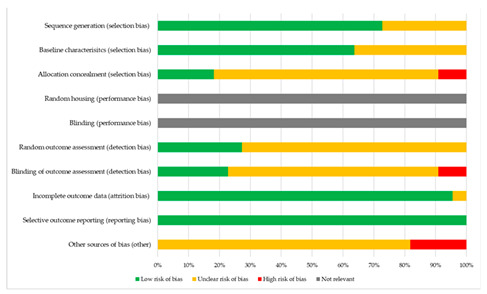

**Table 3 biomolecules-12-00529-t003:** Checklist for the assessment of the methodological quality of human studies.

Study	Reporting	External Validity	Internal Validity—Bias	Internal Validity—Confounding	Power	Total	Grade
Paasche, 2006	9	2	5	2	0	18 (64%)	Fair
Paasche, 2009	9	2	5	2	0	18 (64%)	Fair
Prenzler, 2018	8	2	5	2	1	18 (64%)	Fair
Briggs, 2020	9	2	5	2	1	19 (68%)	Fair

## Data Availability

Data was retrieved using online search engines PubMed, Embase and Cochrane Library.

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
