# Peer review of "Inner Ear Pharmacotherapy for Residual Hearing Preservation in Cochlear Implant Surgery: A Systematic Review"

_biomolecules, 2022, doi:10.3390/biom12040529_

Round 1

Reviewer 1 Report

The manuscript presents review data on  intracochlear administration of corticosteroid and non-corticosteroid agents for residual hearing preservationin coclhear implant surgery. the systematic review is well conducted following the PRISMA guidelines. The data are valid and the conclusions are well supported by the results. 

There is only one point that I prefer to be clarified. In Outcome Measures, line 90, I think it would be better if authors could specify what "impedence" is. Mybe not alle the readers will be audiologist or ENT doctor. For this reason it could be useful to specify that it is an electric impedance evaluation of each electrodes of the cochlear implant and that it is an indirect signe of good coupling between electrode and neural interface.  

Reviewer 2 Report

Overall, this is a well-written and comprehensive review focused on intracochlear drug treatments to preserve residual hearing after cochlear implantation.  The review is inclusive of hearing thresholds, spiral ganglion counts, fibrotic tissue development, and impedance, and does a good job of synthesizing the state of the field.  

Some minor suggestions to improve the manuscript include:

1) Add a table to summarize the main findings of each paper reviewed.

2) Provide some background information on some terms in Tables 2 and 3, specifically sequence generation, baseline characteristics, allocation concealment, reporting, external validity, etc.  The reader will appreciate not having to look up these terms.  

3) 26 studies were reported, but it would also be helpful to know which of the studies came from the same laboratories, i.e. how many unique laboratories are represented here. Many of the studies come from a few labs.

4) In the Introduction, corticosteroids and non-steroidal anti-inflammatory drugs are mentioned, but growth factors and laminin are not described there.  Their sudden appearance in the Results section is therefore unexpected. These should be described briefly in the Introduction along with some background for the rationale for their use (e.g. promote neural health/growth as opposed to suppress inflammation).  Same for the other  drug treaments (e.g. re: breakdown of intracochlear thrombi mentioned in Discussion).

5) In section 3.3.1, what is the difference between DEX and triamcinolone?  What is the rationale for using these two over other available glucocorticoids?

6) The last paragraph of Discussion is particularly useful, and it would be helpful to see some of the various study limitations summarized in a table, or added to Table 1.  The duration of follow-up is already listed, but it would be helpful to see added to a table along with summary results to evaluate which studies to weight more: degree of hearing loss at baseline, implant model (cochleostomy + acute insertion vs long-term implant), duration of drug release, and number of animals used for histology as noted in the limitations.
